



# Using carbon-14 and carbon-13 measurements for source attribution of atmospheric methane in the Athabasca Oil Sands Region

Regina Gonzalez Moguel[1], Felix Vogel[2], Sébastien Ars[2], Hinrich Schaefer[3], Jocelyn C. Turnbull[4,5], Peter M.J. Douglas[1]

[1]Earth and Planetary Sciences Department, McGill University; GEOTOP research center
[2]Environment and Climate Change Canada
[3]National Institute for Water and Atmospheric Research of New Zealand
[4]GNS Science, New Zealand
[5]CIRES, University of Colorado at Boulder, USA

*Correspondence to*: Regina Gonzalez Moguel (regina.gonzalezmoguel@mail.mcgill.ca), Peter Douglas (peter.douglas@mcgill.ca)

## Abstract

The rapidly expanding and energy intensive production from the Canadian oil sands, one of the largest oil reserves globally, accounts for almost 12% of Canada's greenhouse gas emissions according to inventories. Developing approaches for evaluating reported methane ($CH_4$) emission is crucial for developing effective mitigation policies, but only one study has characterized $CH_4$ sources in the Athabasca Oil Sands Region (AOSR). We tested the use of $^{14}C$ and $^{13}C$ carbon isotope measurements in ambient $CH_4$ from the AOSR to estimate source contributions from key regional $CH_4$ sources: (1) tailings

ponds, (2) surface mines and processing facilities, and (3) wetlands. The isotopic signatures of ambient $CH_4$ indicate that the $CH_4$ enrichments measured at the site were mainly influenced by fossil $CH_4$ emissions from surface mining and processing facilities (53 ± 18 %), followed by fossil $CH_4$ emissions from tailings ponds (36 ± 18 %), and to a lesser extent by modern $CH_4$ emissions from wetlands (10 ≤ 1 %). Our results confirm the importance of tailings ponds in regional $CH_4$ emissions and show that this method can successfully separate wetland $CH_4$ emissions. In the future, the isotopic characterization of $CH_4$

sources, and measurements from different seasons and wind directions are needed to provide a better source attribution in the AOSR.





## 1 Introduction


Methane ($CH_4$) is an important greenhouse gas that has 32 times the global warming potential (mass basis) of carbon dioxide ($CO_2$) on a 100-year timescale, and which contributes to the production of ozone, water vapor (in the stratosphere), and $CO_2$ in the atmosphere (Myhre et al., 2013; Etminan et al., 2016). Global $CH_4$ concentration in the atmosphere has almost tripled compared to pre-industrial values (Rubino et al., 2019), largely due to increased anthropogenic activities that include fossil

fuel production and use and agriculture (Jackson et al., 2020; Turner et al., 2019). Since most fossil fuel emissions originate from coal, oil, and natural gas exploitation, transportation, and use (Jackson et al., 2020; Saunois et al., 2020), mitigating $CH_4$ emissions from these activities is necessary to fulfill governmental $CH_4$ emissions reduction goals. Furthermore, a fast $CH_4$ mitigation from the oil and gas sector is projected to have a key role in slowing the rate of global warming over the next few decades (Ocko et al., 2021).

Canada contains approximately 10% of the world's crude oil proven reserves, with 82% of these reserves located in the Athabasca Oil Sands Region (AOSR) in Alberta (Alberta Energy Regulator, 2015). Oil sand deposits, composed of a mixture of sand grains, water, bitumen, and clay minerals (Mossop, 1980; Takamura, 1982), are extracted through two methods. Shallow deposits (< 75 m) are recovered through surface mining and the bitumen is subsequently separated from sands with alkaline warm water, concentrated, upgraded, and refined (Larter and Head, 2014). Residual water, solids, and diluents used

to separate the bitumen are then stored in tailings, which depending on their age and composition emit volatile organic compounds (VOCs), reduced sulfur compounds, $CO_2$, and $CH_4$ (Small et al., 2015). In contrast, the recovery of deeper deposits requires the use of *in situ* techniques that involve lowering the viscosity of bitumen by injecting steam into the reservoir to extract it (Bergerson et al., 2012). Although only around 20% of the oil sands deposits are recoverable using surface mining (Alberta Energy Regulator, 2015), surface mining accounts for 45−65 % of the annual crude oil production from oil sands

(Holly et al., 2016). Each of these methods have greenhouse gas (GHG) emissions associated with them, and it is estimated that the oil sands account for 12% of Canada's total GHG emissions (Environment and Climate Change Canada, 2018). In the AOSR, an aircraft-based study attributed $CH_4$ emissions to three main sources: microbial methanogenesis in tailings ponds (45% of total $CH_4$ emissions), disturbance of mine-faces in open pit mines (50% of total $CH_4$ emissions), and facilities activities such as venting, cogeneration, and natural gas leakage (5% of total emissions) (Baray et al., 2018).

Methane emissions from the oil sands are reported annually to Environment and Climate Change Canada through the Greenhouse Gas Reporting Program (GHGRP), based on inventories of facilities that emit 10 or more kilotonnes of GHG per year (Environment and Climate Change Canada, 2018). The GHGRP and other inventory approaches have varying degrees of accuracy and are vulnerable to uncertainty in the "emission factors" used to calculate the GHG emission rates. Top-down approaches are used to verify inventory-based GHG emission estimates, and aircraft-based top-down estimates in the AOSR

have shown that inventories underestimate GHG emissions (Liggio et al., 2019), with an aircraft-based estimate reporting 48% higher $CH_4$ emissions than in the inventories (Baray et al., 2018). However, these aircraft measurements were limited to a short period of time (summer 2013), and there have not been other studies confirming and updating these findings. Given these





limitations, additional measurements of $CH_4$ and source specific tracers are needed to reconcile differences amongst methods, to generate data at different times of the year, and to generate long-term data for monitoring the evolution of AOSR emissions.

We can use $^{13}C$ and $^{14}C$ carbon isotopes to determine the sources of $CH_4$ emissions because different $CH_4$ sources have distinct isotopic compositions (Sherwood et al., 2017; Whalen et al., 1989). Delta $^{13}C$ ($\delta^{13}C$) denotes the ratio of $^{13}C$ relative to $^{12}C$ compared to the PDB standard and reported in parts per thousand. The $\delta^{13}C$ of $CH_4$ depends on how $CH_4$ is produced: by microbial activity (-61.7 ± 6.2 ‰), by the thermal breakdown of organic molecules (-44.8 ± 10.7 ‰), and by incomplete combustion (-26.2 ± 15 ‰) (Sherwood et al., 2017). Delta $^{14}C$ ($\Delta^{14}C$) reports the ratio of $^{14}C$ relative to $^{12}C$ compared to a

decay-corrected standard and normalized to a $\delta^{13}C$ of -25 per mil to account for fractionation (Stuiver and Polach, 1977). Because fossil fuels lack $^{14}C$, $CH_4$ produced from fossil fuel precursors has a $\Delta^{14}C$ value of -1000‰. In contrast, $CH_4$ produced from other substrates has a $\Delta^{14}C$ signal approximating the contemporary atmospheric $\Delta^{14}CO_2$ value (~5.5 ‰ in 2019) (Whalen et al., 1989; Turnbull et al. 2017). $CH_4$ produced from contemporary substrates do not approximate the atmospheric $\Delta^{14}CH_4$ value, which is enriched due to global nuclear power plant $^{14}CH_4$ emissions (Lassey et al., 2007). The implication is that in the

AOSR, $\delta^{13}C$ can be used to separate thermally produced surface mine and leaking $CH_4$ from the microbially-produced $CH_4$ in tailings ponds, local wetlands, and landfill emissions; and $\Delta^{14}C$ can further separate the fossil microbial $CH_4$ from tailings ponds from the modern microbial $CH_4$ from landfills and wetlands.

Previous studies have shown that $\delta^{13}C$ can be successfully used for $CH_4$ source attribution in urban, natural, and fossil fuel industrial settings (Eisma et al., 1994; Lowry et al., 2001; Fisher et al., 2011; Townsend-Small et al., 2012; Lopez et al., 2017;

Maazallahi et al., 2020), and current instruments allow for relatively cheap and precise $\delta^{13}C$ determinations in small atmospheric samples using gas-source mass spectrometers or cavity ringdown spectrometers. Conversely, $\Delta^{14}C$ measurements have been successful in $CO_2$ source attribution (Lopez et al., 2013; Zimnoch et al., 2012; Turnbull et al., 2015; Miller et al., 2020), but not in $CH_4$ source attribution (Eisma et al., 1994; Townsend-Small et al., 2012). Additionally, $\Delta^{14}C$ measurements are rarely used as analyzing $^{14}C$ requires larger samples than $^{13}C$ analysis, a more demanding extraction of methane from air,

and more expensive measurements using accelerator mass spectrometry. Improvements in the atmospheric methane collection and processing are currently being developed, which could increase the use of $\Delta^{14}CH_4$ measurements in the near future (Zazzeri et al., 2021).

In this study, our main goal is to test the use of combined $\Delta^{14}C$ and $\delta^{13}C$ measurements in ambient $CH_4$ to estimate contributions from the largest $CH_4$ sources in the AOSR region including wetlands, surface mines, and tailings ponds. We expect to provide

a new and practical proof-of-concept method for the long-term monitoring of key $CH_4$ emissions in regions with multiple $CH_4$ sources like the AOSR, which is crucial to developing effective $CH_4$ mitigation policies and, in the specific case study, to fulfill Canada's goal of reducing $CH_4$ emissions from the oil and gas sector by 40–45 % below 2012 levels by 2025 (Government of Canada, 2016).



## 2 Methods

### 2.1 Sampling campaign

The sampling campaign took place between the 13th and 23rd of August 2019 at the Environment Canada atmospheric monitoring site Fort McKay South (FMS), adjacent to the Wood Buffalo Environmental Association Air Monitoring Station 13 (AMS13). The monitoring station is located in the AOSR (57°08'57.54" N, 111°38'32.66" W), surrounded to the East and West by boreal forest and to the North and South by oil sands mining and processing facilities (Figure 1). Air pollution levels at the site depend on the wind direction, and the principal wind directions in Fort McKay are northerly and southerly (Bari and Kindzierski, 2015).

To the North, facilities include the Canadian Natural Resources Limited (CNRL) Horizon Processing Plant and Mine, and Muskeg River and Jackpine Mines; the Fort Hills Oil Sands Mine; Syncrude Aurora North Mine Site; and the Imperial Oil Kearl Processing Plant and Mine (Government of Canada, 2017). $CH_4$ emissions from CNRL Horizon facilities, Muskeg River and Jackpine Mines, and the Syncrude Aurora North Mine have been primarily attributed to open pit mining ($5.2 \pm 1.2$ th$^{-1}$), but significant $CH_4$ emissions originating from the CNRL Horizon main plant facility ($1 \pm 0.3$ th$^{-1}$) have also been detected (Baray et al., 2018). To the South, the main facilities are Syncrude Canada Mildred Lake and Suncor Energy Inc. Oil Sands (Government of Canada, 2017). $CH_4$ emissions from these two facilities have been mainly attributed to tailings ponds ($8.8 \pm 1.1$ th$^{-1}$) followed by open mining ($4.6 \pm 0.6$ th$^{-1}$) (Baray et al., 2018).

We collected air samples in 70 L cylinder tanks by filling the tank to a pressure of 2000 PSI using a Bauer PE-100 compressor with a magnesium perchlorate water trap. We aimed to sample $CH_4$ peaks coming from different wind directions. Before the field campaign, the new Bauer PE-100 compressor was tested at the ECCC laboratories and compared to an existing oil-free RIX compressor system, used to fill reference ('laboratory standards') for ECCC. The difference in methane dry air mole fraction in the cylinders when using the Bauer PE-100 and RIX compressor was found to be within 10 ppb when consecutively filling tanks using ambient air. During the sampling campaign, we flushed the cylinders two times by filling the tank with air until it reached 2000 PSI and subsequently purging the air by opening the tank valve before collecting the air sample.

We performed continuous measurements of methane ($CH_4$), carbon dioxide ($CO_2$), and carbon monoxide (CO) dry air mole fractions for the whole sampling campaign using a Picarro G2401 gas concentration analyzer, which has a five-minute average precision of 1.5 ppb for CO, 20 ppb for $CO_2$, and 0.5 ppb for $CH_4$. From the 13th to the 24th of August, we continuously measured $\delta^{13}CH_4$ using a Picarro G2201-i Isotopic Analyzer, which has a $1\sigma$ precision better than 0.8 ‰ when using the $CH_4$ isotope mode and five-minute averages. However, a data processing error with the Picarro G2201-i allowed us to retrieve only the measurements from the 13th to the 19th of August. The intake lines of all the instruments were attached at the rooftop of the air monitoring station, approximately 3 meters above ground.





## 2.2 CH₄ isotopic analyses

Methane was extracted from the gas samples at the National Institute of Water and Atmospheric Research (NIWA) in Wellington, New Zealand, following the methods described in Lowe et al. (1991), with updates as described in the following. In summary, a mass flow controller set at 1 L/min was connected to the tanks. Air was drawn from the tanks using a 170 L/min rotary pump and pumped through two cryogenic traps to remove $CO_2$, $H_2O$, $N_2O$, and other specific hydrocarbons. Each of these cryogenic traps is made of four 350 mm long loops passing in and out of liquid nitrogen. The loops are made of 12mm

ID Pyrex tubing and are kept at pressures lower than 10 kPa. After these first two traps, the sample passed through a third trap containing a *Sofnocat* reagent (containing platinum and palladium on a tin oxide support) which acts as a catalyst in the conversion of CO to $CO_2$. This $CO_2$ was subsequently removed using two additional cryogenic traps. Next, $CH_4$ was combusted at 750 °C to $CO_2$ and $H_2O$ using an alumina-supported platinum catalyst. The resulting $CO_2$ was collected and purified in three additional cryogenic traps. Last, $H_2O$ was removed using alcohol dry ice traps at -80 °C and $CO_2$ was vacuum distilled into

glass vials or break seals for mass spectrometry. Separate extractions were carried out for each $^{13}C$ and $^{14}C$ analysis, processing 26 L of air for $^{13}C$ and 230-290 L for $^{14}C$ (depending on $CH_4$ content of the sample), respectively.

Analysis of $^{13}C$ was performed on a Thermo MAT-253 isotope ratio mass spectrometer (IRMS) in dual inlet mode. Samples were analyzed against a pure $CO_2$ working reference gas derived from a $^{13}C$ depleted barium carbonate standard (NZCH). The standard deviation for a $\delta^{13}C$ determination is 0.02 ‰. The results were reported relative to PDB-$CO_2$. For $^{14}C$ analysis, the

methane-derived $CO_2$ was reduced to graphite using $H_2$ and an iron catalyst at 550 °C (Turnbull et al., 2015) and measured for $^{14}C$ content by accelerator mass spectrometry (Zondervan et al., 2015). The results were reported as fraction modern carbon and $\Delta^{14}C$ age corrected to date of sample collection following internationally agreed conventions (Stuiver and Polach, 1977, Donahue et al., 1990, Reimer et al., 2004). The measurement precision for this dataset is 2.2 to 2.6 ‰ in $\Delta^{14}C$.

## 2.3 Back-trajectory modelling using HYSPLIT-5

We generated hourly 12-hour backward trajectories for the duration of the sampling campaign using HYSPLIT-5. HYSPLIT is a model for computing atmospheric transport and dispersion of air masses developed by NOAA's Air Resources Laboratory, and a more complete description of the system can be found in Stein et al. (2015). In this model, a back-trajectory is calculated from a particle that represents a gas being moved by the mean wind field. To calculate the concentration of the trace gas (air concentrations), a number of particles are released from the receptor and dispersion equations are applied to the upwind

trajectory calculation. Then, the mass of the computed particles is added and divided by the volume of their horizontal and vertical distribution. In both cases we configured the model to start at 50 m above ground level from the location of the FMS site and to use meteorological parameters from the NAM 12-km (hybrid sigma pressure US 2010-Present) database.





## 2.4 Estimating source contributions using keeling plots

The Keeling plot approach is based on the conservation of mass in the lower planetary boundary layer (Keeling 1958; 1961). It assumes that the atmospheric $CH_4$ is the result of a simple linear mixing between two components, background $CH_4$ and the sum of all $CH_4$ sources, and that the isotope ratio of the two components does not change substantially over time, as in this study. As a result, the intercept of a simple linear regression between $1/[CH_4]$ and $\delta^{13}CH_4$ or $\Delta^{14}CH_4$ from atmospheric samples is interpreted as the mean isotopic signature of the $CH_4$ sources (Eq. 1 and 2).

$$\Delta^{14}C_{air} = \frac{C_{background}(\Delta^{14}C_{background}-\Delta^{14}C_{source})}{C_{air}} + \Delta^{14}C_{source} \tag{1}$$

$$\delta^{13}C_{air} = \frac{C_{background}(\delta^{13}C_{background}-\delta^{13}C_{source})}{C_{air}} + \delta^{13}C_{source} \tag{2}$$

Because the source isotopic signature represents the weighted sum of all the $CH_4$ sources, a mixing model (a mass balance) can be used to determine the individual $CH_4$ source contributions from the mean $CH_4$ source isotopic signature if the individual source isotope signatures are known. We used MixSIAR, a Bayesian isotope mixing model framework implemented as an open-source R package, to estimate the contribution of potential $CH_4$ sources to the 'mixture mean' (mean source signature in air samples). The main difference between Bayesian mixing models and simple mixing models is that the former considers the uncertainty in the isotopic source values (Stock et al., 2018). To account for source uncertainty, the user provides summary statistics of the source isotopic values (mean, variance, and sample size) and source parameters are fitted as in Ward et al. (2010). The source isotopic values used in the mixing model were derived from the literature and are described in the following section.

## 3 Results and discussion

### 3.1 Isotopic signature of CH₄ sources in the AOSR

To estimate the proportion of $CH_4$ emitted from different potential sources, the isotopic signatures of these potential sources must be known. However, specifying the $\delta^{13}CH_4$ from these sources can be especially challenging because $\delta^{13}CH_4$ signatures can have wide ranges and vary locally (Sherwood et al., 2017), and there are no studies isotopically characterizing $CH_4$ from different sources in the AOSR. Based on the previous aircraft source attribution study (Baray et al., 2018), we identified two main $CH_4$ source categories: $CH_4$ emissions related to the mining and processing of bitumen (e.g., leaking and venting), and tailings ponds $CH_4$ emissions. Thermogenic $CH_4$ associated with Alberta's Lower cretaceous oils varies between -42 and -48‰ (Jha, Gray and Strausz, 1979; Tilley et al., 2007), but the prevalence of anaerobic biodegradation in shallow subsurface petroleum reservoirs changes the $\delta^{13}CH_4$ composition of heavily degraded oils to between -45 to -55 ‰, in particular by hydrogenotrophic $CH_4$ production (Head, Jones, Larter, 2003; Jones et al., 2008). This biogenically over-printed thermogenic $CH_4$ is present in the mined material of the AOSR, which is potentially released when oil sands are mined, but also during





transport, ore preparation, and extraction of bitumen (Johnson et al., 2016). Thus, we used this $\delta^{13}C$ range to represent $CH_4$ emissions derived from the bitumen mining and processing (Table 1).

Residual water generated from the surface mining process is stored in tailings ponds where aerobic and anaerobic degradation
are mainly fueled by certain naphtha components in the diluents, in specific short-chain n-alkanes ($C_6$ to $C_{10}$), BTEX compounds (i.e., toluene and xylenes), and long-chain n-alkanes ($C_{14}$ to $C_{18}$) (Siddique et al., 2006, 2007, 2011, 2012). Radiocarbon measurements of tailings ponds components, including total organic carbon (TOC), total lipid extract (TLE), and phospholipid fatty acids (PLFAs) have yielded $\Delta^{14}C$ signatures of approximately -995 ‰ (Ahad and Pakdel, 2013). We infer that $CH_4$ is most likely produced from these substrates and therefore has the same $\Delta^{14}C$ signature (Table 1). The chemical
composition of the tailings ponds − determined by mineralogy of the oil sands, extraction techniques and additives used, and age of the ponds− influences the microbial communities involved in the substrate degradation (Small et al., 2015), which are likely dominated by syntrophic communities as well as both acetoclastic methanogens, previously associated to short n-alkane degradation, and hydrogenotrophic methanogens, associated to the metabolism of long-chain alkanes and BTEX (Penner and Fought, 2010; Shahimin et al., 2016; Siddique et al., 2012; Zhou et al., 2012). Measurements of the dissolved $\delta^{13}CH_4$ from the
hypolimnion of Base Mine Lake, a dimictic end pit lake, ranges between -60 and -65 ‰ and to our knowledge are the only available $\delta^{13}CH_4$ measurements associated to oil sands lakes (Goad 2017). However, variations in the microbial community composition between ponds results in variations in the rate of $CH_4$ production (Small et al., 2015), and might also result in differences in the $\delta^{13}CH_4$ due to different fractionation in acetoclastic and hydrogenotrophic methanogenesis (Whiticar 1999; Whiticar, Faber, and Schoell, 1986). Moreover, the Base Mine Lake $\delta^{13}CH_4$ value should be regarded as a minimum, because
methanotrophic communities are active in the surface of the tailings, most likely shifting the $\delta^{13}CH_4$ towards more positive values during partial oxidation of methane before emission to the atmosphere (Saidi-Mehrabad et al., 2013).

We identified three additional regional $CH_4$ sources not directly originating from to the oil sands mining and processing facilities but that could be contributing to $CH_4$ emissions in the region. First, landfills located in Fort McMurray, as some of the back trajectories show air masses coming from the general Fort McMurray direction. Second, boreal wetland $CH_4$
emissions, estimated to have a mean $\delta^{13}C$ value -67.8 ‰, based on atmospheric measurements (Ganesan et al., 2018), and a predominantly modern $\Delta^{14}C$ signature, even if there are wetlands associated to permafrost collapse in the region that might emit some pre-modern $CH_4$ (Cooper et al., 2017; Estop-Aragonés et al., 2020). The third potential source of $CH_4$ are forest fires, as three major wildfire events occurred in 2019 in Alberta: the Battle complex (Peace River area), Chuckegg Creek wildfire (High Level area), and the McMillan complex (Slave Lake area). The three events started in May and were declared
under control the 26[th] of June, 1[st] of July, and the 18[th] of August, respectively (MNP LLP 2020), with the third event briefly overlapping with some of the sampling dates (16[th] to 18[th] of August). However, the event was 290 km Southwest of the sampling site, while the air in the sampling site originated from the Northwest (see section 3.2), and therefore it is unlikely that this was a significant source of $CH_4$ in the air samples. From the three sources, we speculated that wetlands are the most prominent $CH_4$ source since at a provincial level (Alberta), $CH_4$ wetland emissions are estimated to be 2.5 to 3.5 Tg a$^{-1}$, roughly



half of the total anthropogenic emissions in the province (Baray et al., 2021). Therefore, we only considered wetland emissions in this category to avoid having an underdetermined mixing model. If we were to add a landfill component, assuming a $\delta^{13}$C value of -55 for landfills (Lopez et al., 2017), the estimation would result in a larger contribution of microbial fossil $CH_4$ relative to thermogenic $CH_4$. For example, if 10% of the microbial modern emissions were derived from landfills and 90% from wetlands, our model estimate of the contribution from tailings ponds increases by 2% (See Sect. 3.3).


### 3.2 Isotopic signature of ambient $CH_4$

Analyses of the 12-hour back trajectories for the 11-day sampling campaign showed that air masses arriving at the FMS station during this time period primarily originated from three general directions (Figure 2B): from the West between the 13 and 15 of August, from the Northwest between the 16 to 19 of August, and from the Southwest and Southeast between the 20 to 24

of August. The $CH_4$ concentration time series for this time period indicated that most $CH_4$ enrichments were associated to trajectories originating from the West and South, in particular from air masses that transit over the Syncrude Mildred Lake facilities and CNRL Horizon oil sands facilities (Figure 2). The collection of air samples corresponded to the period from the 16 to the 24 of August, and therefore the collected air originated from two main directions, the Southwest – Southeast and the North.

The $CH_4$ concentration, $\delta^{13}CH_4$, and $\Delta^{14}CH_4$ of the air samples are shown in Table 1. There were significant correlations between $1/[CH_4]$ and $\Delta^{14}CH_4$ ($r^2 = 0.99$, $p < 0.05$), between $1/[CH_4]$ and $\delta^{13}CH_4$ ($r^2 = 0.84$, $p < 0.05$), and between $\Delta^{14}CH_4$ and $\delta^{13}CH_4$ ($r^2 = 0.8$, $p < 0.05$) in the air samples associated to back-trajectories originating from the South and Southwest, corresponding to August $20^{th}$ to $24^{th}$ (black lines in Figure 3). The intercept of the $\Delta^{14}C$ Keeling plot for these samples showed a source signature of -893 ± 53 ‰ (Figure 3A), while the intercept of the $\delta^{13}C$ Keeling plot yielded a source value of -56 ± 1.2

‰ (Figure 3B).

Samples associated to back-trajectories originating from the North, corresponding to August $16^{th}$ to $19^{th}$, had only one datapoint with a higher $\delta^{13}C$ than background air. When building a Keeling plot with these five samples, the $\delta^{13}C$ intercept showed a source value of – 35 ± 4 ‰ (red lines in Figure 3B), which could point to a thermogenic or pyrogenic source of $CH_4$ originating in the Northern mines.

### 3.3 Source contributions

The approximate contributions from each source category to samples associated with back-trajectories originating from the south were calculated with MixSIAR and are shown in Figure 4. The microbial and thermogenic fossil enrichment observed in the $CH_4$ air samples (~90%), indicate that most of the $CH_4$ enrichment observed at the site was influenced by $CH_4$ emissions from the oil sands mines and processing facilities. Specifically, the contribution from thermogenic $CH_4$ was estimated to be

53 ± 18 % while the contribution from fossil microbial $CH_4$ from tailings ponds contribution was estimated to 36 ± 18 %,



although a large uncertainty was associated to both estimates (Figure 4B). The results also indicate an influence of approximately $10 \le 1\%$ from microbial modern sources (Figure 4b), most likely from wetlands. If most of the microbial modern enrichment is derived from wetlands, it is likely that the contribution from this source is near the annual maximum, as $CH_4$ wetland emissions typically peak in the summer (Baray et al., 2021).

Analyses of the back-trajectories indicated that the air masses from which these sample were collected originated from the south, and therefore the samples are likely predominantly influenced by the Syncrude and Suncor facilities and tailings ponds. This would explain the substantial enrichment of fossil microbial $CH_4$ in our samples, as measurements of $CH_4$ emissions have shown that the largest $CH_4$ emitting tailings management areas are Syncrude's Mildred Lake Settling Basin and the Base Mine Lake (Small et al., 2015; You et al., 2021). In comparison to the oil sands facilities in the south (Syncrude Mildred Lake and

Suncor), the facilities to the North of the air monitoring site have been shown to have much larger $CH_4$ contributions from surface mining and natural gas leaking and venting (Baray et al., 2018), as tailings ponds emissions are minimal (below 1 ton/ha/year) (Small et al., 2015). This was reflected in the few air samples originating from the north that show an enriched $\delta^{13}CH_4$, consistent with the isotopic signature of thermogenic $CH_4$ (Figure 3B).

Compared to the only previous $CH_4$ source attribution study available (Baray et al., 2018), our results implied a lower

contribution from tailings ponds and a larger contribution from surface mines and processing facilities. Baray et al. (2018) estimated that 65% of $CH_4$ emissions from the Syncrude Mildred Lake and Suncor mines and facilities originated from tailings ponds and 34% from surface mines, but there have not been studies updating these estimates since this study was performed in summer 2013. While it is likely that the emissions have changed since 2013, we suggest that the difference between studies are partly a result of the large uncertainty of our estimates. This uncertainty in the estimate is mainly due to the uncertainty in

the isotopic signatures of $CH_4$ sources. For example, a change of 5 ‰ towards more positive values in the tailings ponds $\delta^{13}CH_4$ signature due to microbial oxidation of $CH_4$ in the epilimnion, would increase the calculated contribution from tailings ponds to $51 \pm 21$ % and decrease the thermogenic contribution to $40 \pm 21$ %. This example illustrates the need to reduce the uncertainty in the source isotopic signatures with an extensive $\delta^{13}C$ characterization of $CH_4$ sources in the AOSR, in particular from tailings ponds and surface mines. Furthermore, the use of additional tracers such as methane/ethane ($C_2H_6/CH_4$) ratios

and $\delta^2H$ in $CH_4$ could help constraining emissions from source categories since biogenic and thermogenic processes yield distinctive $CH_4/C_2H_6$ ratios and $\delta^2H$ in $CH_4$ (Townsend-Small et al., 2016; Lopez et al., 2017; Douglas et al., 2021).

While an exhaustive $\delta^{13}C$ characterization of $CH_4$ sources is needed to improve source estimates using carbon isotopes, the clear correlations in our air samples show that this method is useful for estimating $CH_4$ source contributions in regions with multiple $CH_4$ sources like the AOSR. Moreover, the collection of air in cylinders is less costly and easier to do on a regular

basis compared to techniques such as aircraft measurements and therefore is well suited for monitoring how source emissions change with time (seasonally and annually). The use of an instrument for continuous $\delta^{13}CH_4$ measurement such as a Picarro G2201-I Isotope Analyzer could make this process even easier and more evenly distributed through the year. Although our results of the 6-day hourly-averaged $\delta^{13}CH_4$ measurements are not shown here due to problems with the instrument calibration,





we did observe a clear linear relationship ($p < 0.05$) between the hourly $1/CH_4$ and the hourly-averaged $\delta^{13}CH_4$ on two of the six sampling days, which corresponded to when the air masses originated from the West.

## 4 Summary and conclusions

We conducted a sampling campaign in the Athabasca Oil Sands Region in summer 2019 with the objective of evaluating the potential of using combined $\Delta^{14}C$ and $\delta^{13}C$ measurements in ambient $CH_4$ for source attribution. We demonstrated the use of this method for separating emissions from three sources: mines and processing facilities, tailings ponds, and regional wetlands. Our results confirm the importance of tailings ponds in regional $CH_4$ emissions (Baray et al., 2018), which we estimated to be approximately 36 % of all the emissions in the region. Furthermore, the addition of $\Delta^{14}C$ in the measurements allowed us to separate wetland $CH_4$ emissions, which are a major provincial source of $CH_4$ (Baray et al., 2021) and therefore have the potential to interfere in the accuracy of top down $CH_4$ estimates. In general, this method showed to be a suitable tool for $CH_4$ source attribution in the AOSR and potentially other oil producing regions as there are clear correlations between between $\delta^{13}C$ and $\Delta^{14}C$, isotopic measurements are cheap relative to other approaches such as aircraft measurements, and the instrumentation set-up allows for continuous year-round measurements.

Although this study is the first to provide a conclusive source attribution using combined $\Delta^{14}C$ and $\delta^{13}C$ measurements in ambient $CH_4$, there are still large uncertainties associated with this method, mainly due to the lack of $\delta^{13}C$ data from key $CH_4$ sources. These uncertainties can be addressed with a characterization of $\delta^{13}C$ and $\Delta^{14}C$ in the main $CH_4$ sources and using additional tracers such as methane-ethane ratios and $\delta^2H$ signatures. Moreover, future work should focus in adding measurements at different times of the year and in consecutive years, as seasonal and annual variations in $CH_4$ emissions are currently not well constrained. At a seasonal scale, temperature changes in the winter probable reduce microbial methanogenesis, decreasing tailings ponds and wetlands emissions, and snow cover in open mining areas could affect $CH_4$ emissions. At an annual scale, changes in mine and processing facilities operations, the development of in-situ mining over surface mining, and changes in the age-dependent tailings pond emission profile could also result in $CH_4$ emission variations. Consequently, implementing isotopic measurements for long term $CH_4$ emission monitoring is essential to have a complete understanding of $CH_4$ emissions in the AOSR and for developing effective mitigation policies.

## 5 Acknowledgements

We thank Lauriant Giroux for the compressor testing and support in the field; Tony Bromley, Sally Gray, Rowena Moss and Ross Martin for sample processing, GC and IRMS analyses; the Rafter Radiocarbon Lab team for [14]C analyses; and Ralf Staebler and Doug Worthy for the ECCC internal review of the manuscript.





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



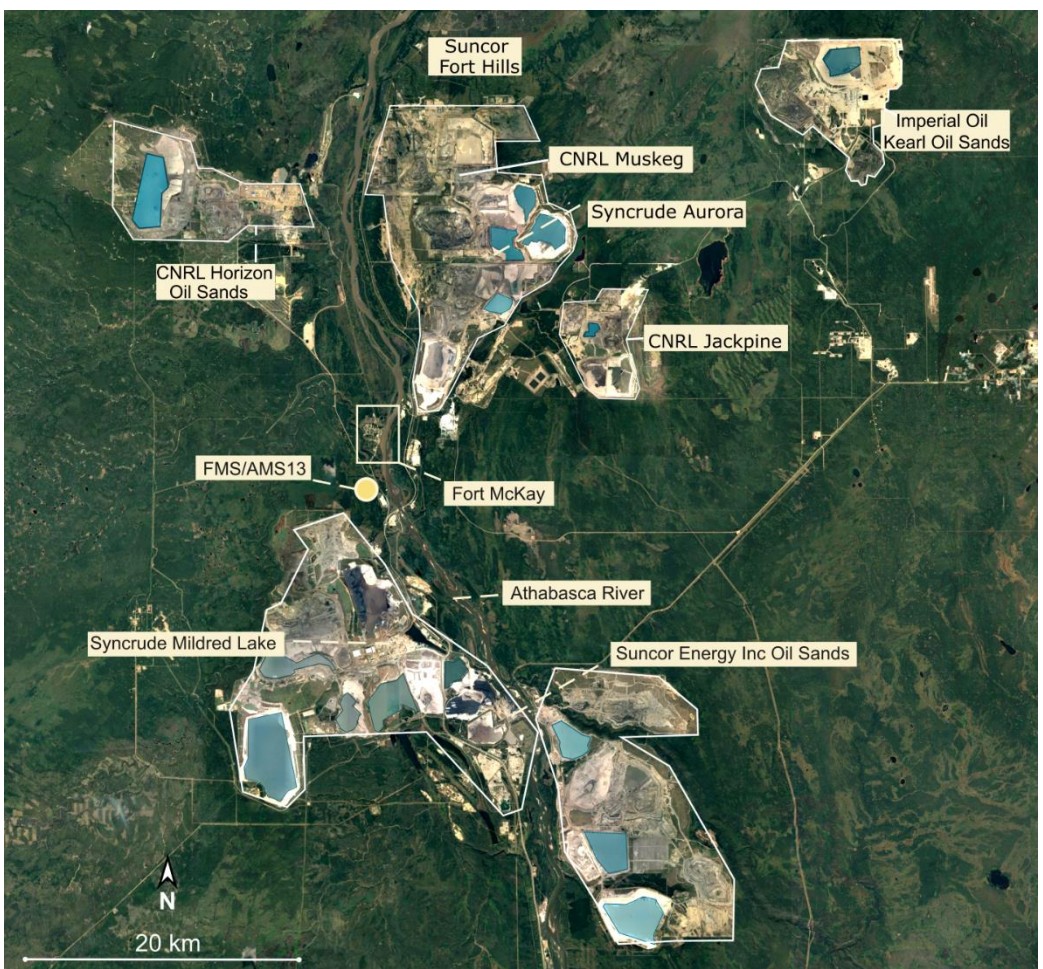

**Figure 1. Satellite view of the Athabasca Oil Sands Region (from © Google Earth) showing the location of oil sands mining and processing facilities and the FMS/AMS13 site from which samples described in this paper were collected (57°08'57.54" N, 111°38'32.66" W).**

**Table 2.**

**Estimated values of $\delta^{13}CH_4$ and $\Delta^{14}CH_4$ for the three source categories used in the source attribution.**

| Source Category | Potential Sources | Estimated $\delta^{13}C$ | Estimated $\Delta^{14}C$ |
|---|---|---|---|
| Thermogenic Fossil | Surface mining, extraction and upgrade, venting, leaking | -45 to -55 ‰[a] | -1000 ‰ |



| Microbial Fossil | Tailings ponds | -60 to -65 ‰[b] | -995 to -1000 ‰[d] |
| Microbial Modern | Canadian boreal wetlands | - 65 to -68 ‰[c] | -3 to 9 ‰[e] |

(a) $\delta^{13}CH_4$ associated to heavily degraded oils from Head, Jones, and Larter (2003)

(b) Hypolimnetic $\delta^{13}CH_4$ values from Base Mine Lake from Goad (2017)

(c) Canadian boreal wetlands $\delta^{13}CH_4$ from Ganesan et al. (2018)

(d) Tailing pond substrate signature from Ahdal and Pakdel (2013)

(e) August 2019 atmospheric $\Delta^{14}CO_2$ extrapolated from Turnbull et al. (2017)

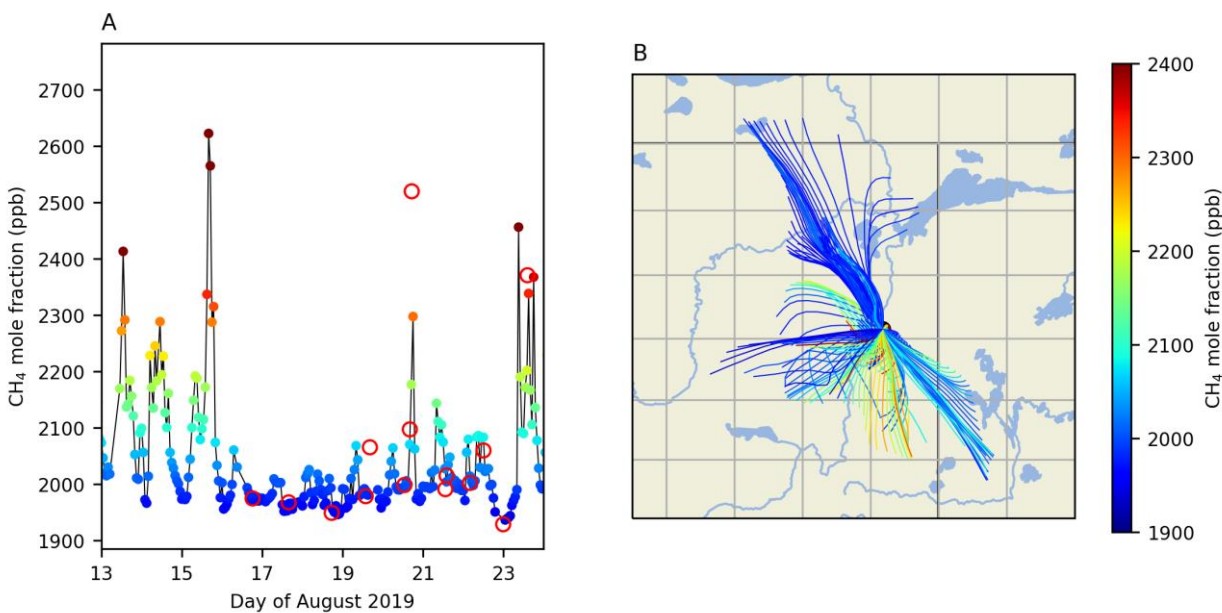


**Figure 2. A) Hourly CH₄ dry air mole fraction measurements at the FMS13 station (Fort McKay South), with the CH₄ concentration of the collected air samples in red circles. B) HYSPLIT 12-hour back-trajectories associated with hourly measurements with color scale representing CH₄ dry air mole fractions in both panels.**

**Table 1.**

**Methane mole fraction, $\delta^{13}CH_4$, and $\Delta^{14}CH_4$ of air samples collected in the Athabasca Oil Sands Region in August 2019.**

| Sample | Date and time (UTC) | CH₄ (ppb) | $\delta^{13}CH_4$ (‰) | $\Delta^{14}CH_4$ (‰) |
| --- | --- | --- | --- | --- |





| 1 | 16/08/2019 18:14 | 1974.5 | -48.1 ± 0.2 | 336.4 ± 2.6 |
| 2 | 17/08/2019 15:46 | 1967.5 | -48.1 ± 0.2 | 337.0 ± 2.6 |
| 3 | 18/08/2019 17:28 | 1948.8 | -48.0 ± 0.2 | 349.8 ± 2.6 |
| 4 | 19/08/2019 13:46 | 1978.4 | -48.2 ± 0.2 | 346.4 ± 2.6 |
| 5 | 19/08/2019 16:16 | 2065.3 | -47.4 ± 0.2 | 275.7 ± 2.5 |
| 6 | 20/08/2019 12:50 | 1998.2 | -48.4 ± 0.2 | 341.2 ± 2.6 |
| 7 | 20/08/2019 16:05 | 2097.1 | -48.9 ± 0.2 | 281.9 ± 2.5 |
| 8 | 20/08/2019 17:14 | 2520.0 | -50.2 ± 0.2 | 68.5 ± 2.2 |
| 9 | 21/08/2019 13:17 | 1990.9 | -48.4 ± 0.2 | 333.7 ± 2.6 |
| 10 | 21/08/2019 14:00 | 2015.2 | -48.5 ± 0.2 | 315.8 ± 2.6 |
| 11 | 21/08/2019 3:55 | 2002.0 | -48.0 ± 0.2 | 325.1 ± 2.6 |
| 12 | 22/08/2019 12:04 | 2059.7 | -48.7 ± 0.2 | 299.5 ± 2.6 |
| 13 | 22/08/2019 23:49 | 1928.6 | -47.9 ± 0.2 | 345.4 ± 2.6 |
| 14 | 23/08/2019 14:19 | 2370.9 | -49.0 ± 0.2 | 132.3 ± 2.4 |

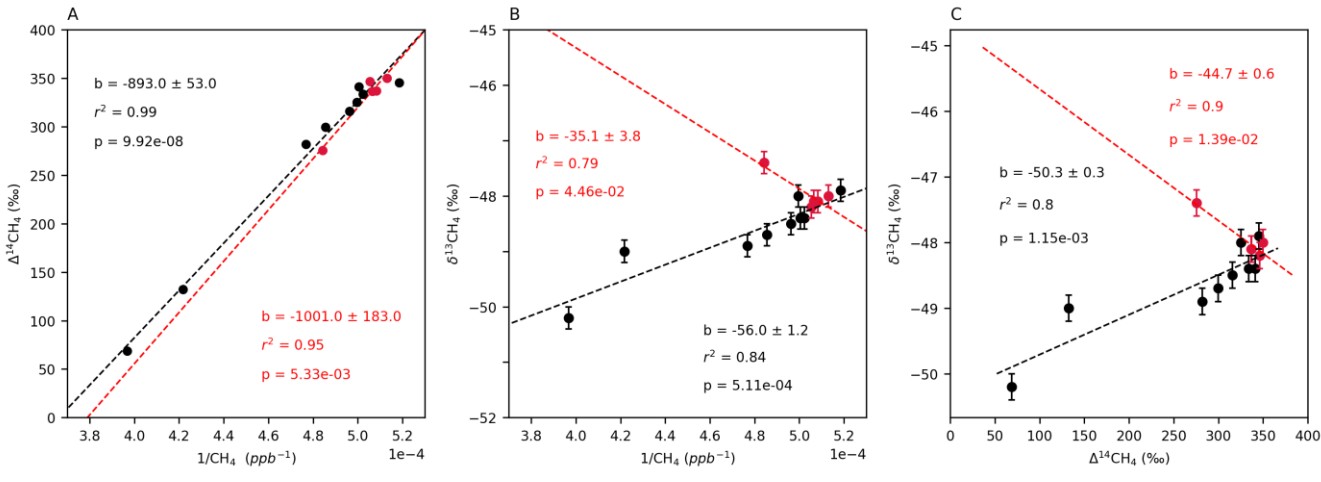






**Figure 3. Keeling plots of: (A) CH₄ and Δ¹⁴CH₄, (B) CH₄ and δ¹³CH₄, and (C) plot of δ¹³CH₄ and Δ¹⁴CH₄ in air samples collected from the 20th to the 24th of August (South) in black (n = 9) and from the 16th to the 20th of August (North) in red (n = 5). In panels A and B, *b* is the intercept of the Keeling plot.**

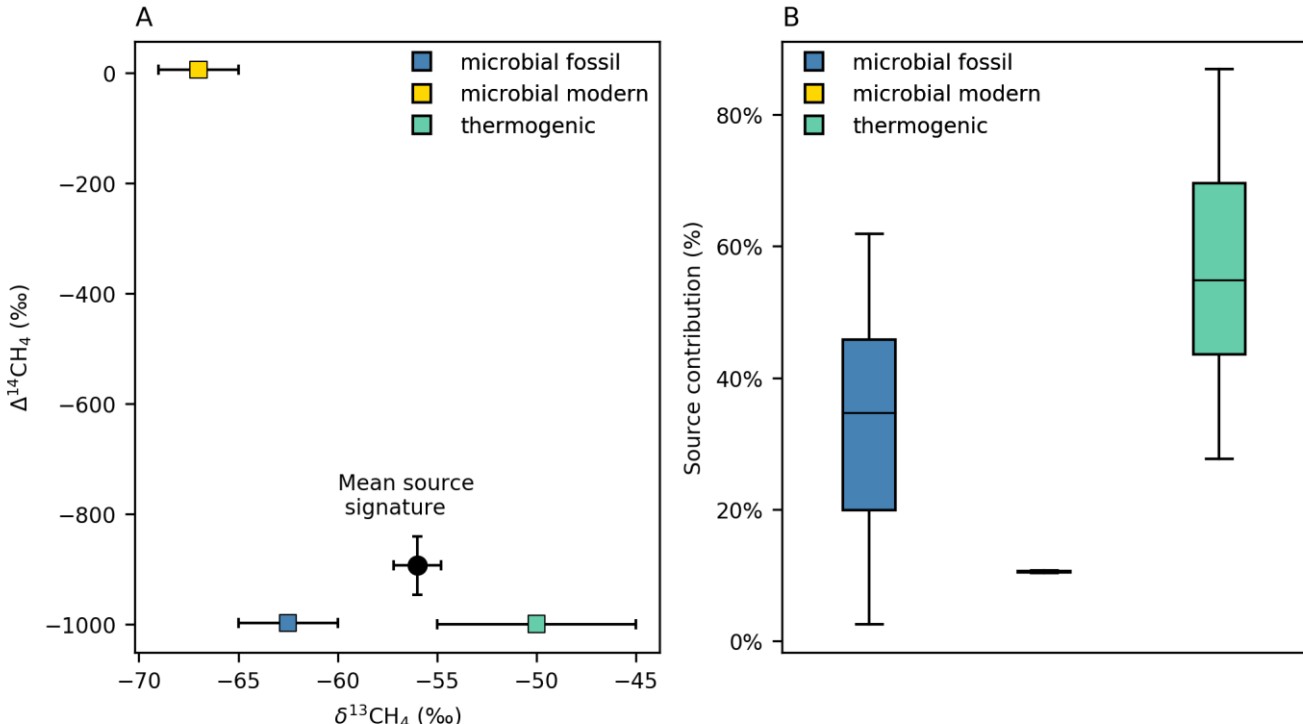


**Figure 4: (A) δ¹³C and Δ¹⁴C signatures of potential CH₄ sources used to estimate source contribution using MixSIAR and mean δ¹³CH₄ and Δ¹⁴CH₄ source signatures derived from Keeling plots (B) Boxplot of the estimated source contributions from microbial fossil CH₄ (tailing ponds), thermogenic CH₄ (surface mines and others), and microbial modern CH₄ (wetlands). The line inside the boxes represents the median, boxes indicate the 25th and 75th percentiles, and whiskers the 5th and 95th percentiles.**
