# Peer review of "Using carbon-14 and carbon-13 measurements for source attribution of atmospheric methane in the Athabasca Oil Sands Region"

_Atmospheric Chemistry and Physics, 2021_

## Author Response (AR1)

On behalf of my coauthors, I thank the reviewers for taking the time to carefully read the manuscript and for the positive comments they provided. We propose the following major changes to incorporate the reviewers' suggestions and improve the manuscript:

1. HYSPLIT trajectories. There was a mistake in the text that said that HYSPLIT trajectories were done at 50 m but the figure showed the 3 m trajectories. Originally, we had ran trajectories at 3m, 10 m, and 50 m, and were planning to show the 50 m trajectories because they minimize the interaction of trajectories with the underlying terrain, but the topography of our study site is flat and this was likely not an issue. We corrected this issue in the manuscript text.

2. We excluded the Picarro isotope data from the results, as it is not used to draw the conclusions, and since the atmospheric data was taken from after the 16 of August, we changed the length of the campaign showed in the results.

3. We corrected the labelling for Tables 1 and 2, added and corrected the uncertainties of the values, and added wind directions.

4. In the original manuscript we calculated the fitting parameters (slope and intercept) of the Keeling Plot using the Ordinary Least Squares (OLS) method, and we obtained their uncertainty by computing the standard deviation from the covariance matrix. The OLS method does not account for errors in variables, and therefore these are not included in the uncertainty of the fitting parameters. To correct for this issue and include the measurement error we changed the regression method from OLS to the York fit (York et al., 2004). The Keeling plot intercepts changed from $\Delta^{14}C$ = -893 ± 53 ‰ and $\delta^{13}C$ = -56 ± 1.2 ‰ to $\Delta^{14}C$ = -898 ± 9 ‰ and $\delta^{13}C$ = -56 ± 0.8 ‰ (for the period from the 16[th] to the 20[th] of August) and from $\Delta^{14}C$ = -1001 ± 183 ‰ and $\delta^{13}C$ = -35.1 ± 3.8 ‰ to $\Delta^{14}C$ = -1007 ± 58 ‰ and $\delta^{13}C$ = -35.1 ± 4.5 ‰ (for the period from the 20[th] to the 23[rd] of August; note that values below -1000 ‰ are not possible, so these values are -1000 ‰). These changes are small and don't affect the wider results and the conclusion of the manuscript.

5. We are now using a dataset from the Northern Hemisphere for the contemporary atmospheric value of $\Delta^{14}CO_2$ because the value we were using was extrapolated from a dataset from the Southern Hemisphere. Second, we used a wider range on values (from 2009 to 2019) because the average residence time of carbon in wetlands is likely to be on the decadal timescales, which means that wetland $CH_4$ carbon could be enriched in $\Delta^{14}CO_2$ relative to contemporary atmospheric $CO_2$.

**Response 1**

1. "Provide a reference for the value of contemporary atmospheric $\Delta^{14}CO_2$ value of ~5.5 ‰ in 2019. Shouldn't now approximate 0 ‰?" The range of $\Delta^{14}CO_2$ values for August 2019 was extrapolated from the dataset in Turnbull et al., 2017. However, since this value corresponds to the Southern Hemisphere, we have updated the calculations using a dataset from Germany and Switzerland (Hammer and Levin, 2017). The resulting value averaged – 5 ‰.

2. "Indicate a value for the current atmospheric $\Delta^{14}CH_4$." Providing a value for the current atmospheric $\Delta^{14}CH_4$ is not completely straightforward, as there are currently not published databases for $\Delta^{14}CH_4$ in the Northern Hemisphere. The only estimate we found in the literature

was 350 ‰ for 2009 (Graven, Hocking, and Zazzieri, 2019). $\Delta^{14}CH_4$ measurements from baring head (New Zealand, Southern Hemisphere) from that same year was 331 ± 4 ‰. If we assume the difference between these two values defines the interhemispheric gradient (~20 ‰), the contemporary $\Delta^{14}CH_4$ for the Northern Hemisphere would be around 340 ‰, as the 2019 value from Baring Head was 322 ± 4 ‰. We added this value in line 74: $CH_4$ produced from contemporary substrates do not approximate the atmospheric $\Delta^{14}CH_4$ value (estimated to be around 340 ‰), which is enriched due to global nuclear power plant $^{14}CH_4$ emissions (Lassey et al., 2007).

3. "Specify why $^{14}C$ measurements in atmospheric $CH_4$ are difficult. You should mention not only the sampling challenges but also the difficulties in quantifying the influence of emissions from nuclear power plants." Emissions from nuclear power plants are a source of uncertainty for global attribution of methane sources, as their $^{14}CH_4$ emissions are poorly constrained. At a regional/local scale, this can also be a source of uncertainty in places such as continental Europe (Eisma et al, 1995) where there is a strong influence from nuclear plants $^{14}CH_4$ emissions, and this nuclear source represents an additional $CH_4$ to the CH4 "mixture". In other sites such as our site, nuclear emissions do not represent a substantial complication at the regional scale.

4. "How long does it take to fill a cylinder?" Around 10 minutes, we changed line 113 to make this clearer: We collected air samples in 70 L cylinder tanks by filling the tank for around ten minutes to a pressure of 13.8 MPa using a Bauer PE-100 compressor with a magnesium perchlorate water trap.

5. "120 An average time of 20 min might reduce the uncertainty in the Picarro isotopic measurements. Did you measure the Allan Variance?" We originally used 1-hour averages to make it comparable to the concentration measurements and we didn't measure the Allan Variance. In the current version of the manuscript, we removed the Picarro isotopic data.

6. "151 Why 50 m above the ground?" See change 1 discussed above.

7. "Table 2: Is it actually table 1?" Yes, the tables were misplaced, and the reference to the tables mislabeled. This has now been corrected.

8. "202 It would be nice to see in Figure 1 the location of the landfill and wetlands as well" We changed the area extent of the map to include the landfills and wetland approximate area in the figure. See the new Figure 1.

9. "Table 2 Can you add a column with the wind direction or the air provenance for each sample?" We have added add a column with the general wind direction for each sample, see the new Table 1.

10. "How did you calculate the uncertainty on the intercept of the Keeling plot?" See change 4 above.

11. "Session 3.3 How does MixSIAR work? I think you should add a reference here (e.g. author of the R package) so that the reader can look at the statistics behind".

It is a bit complicated to explain how MixSIAR works in the text because the models and methods used depend on what information the user provides, and the nature of the system the user is studying. The base of the framework is a mixing model in which the tracer value of the mixture (e.g. $\delta^{13}C_{mixture}$) is the sum of the mean value of the tracer in the source multiplied by their proportional contribution to the mixture as in Eq. 1.

$$\delta^{13}C^{mixture} = \sum_k \delta^{13}C_k^{source}p_k \quad \text{Eq.1}$$

The assumptions for this model are that all the sources are known , tracers are conserved through the mixing process, tracer values are fixed, the tracer values differ between sources, and that the sum of the proportional contributions (*p*) is 1 (Stock et al. 2018).

MixSIAR then incorporates error structures to represent the variability in tracer values of the mixture. The method to model this variability depends on the parameters that the user provides. In our case, for example, because we provided a mean mixture value for each tracer (one source value for $\delta^{13}C$ and one for $\Delta^{14}C$), MixSIAR assumes that the mixture value is a normal distribution defined by the mean of the tracer value, with a variance generated by a weighted combination of source variances (the $\delta^{13}C$ and $\Delta^{14}C$ mean values and uncertainties for each of the three sources we specified tailings ponds, surface mining, wetlands).

We added a short description of how MixSIAR works (see section 2.4) and we also added the reference for MixSIAR in line 166: We used MixSIAR, a Bayesian isotope mixing model framework implemented as an open-source R package (see Stock et al., 2018)

12. "247 10 £ 1%?" This is a typo. The corrected sentence now reads: The results also indicate an influence of approximately 10 ± < 1% from microbial modern sources (Figure 4b), most likely from wetlands.

13. "278 In session 2.1 you say that a data processing error with the Picarro G2201-i allowed to retrieve only the measurements from the 13th to the 19th of August. and you mention a clear linear relationship. I think a plot showing that should be included in the manuscript as well. You could add these measurements in the supplementary material?" See change 2 above, we have removed the sections of the Picarro G2201-I in the revised manuscript.

**Reviewer 2**

1. "The HYSPLIT back-trajectories used were generated for a height of 50 metres above ground level (line 151). Back-trajectories at this height may not correspond to the back-trajectories at a height of 3 m, where measurements were made. The meteorology could be different between these heights and so will the back-trajectories. As the results rely heavily on the back-trajectories to evaluate the source origin, back-trajectories should be generated at a height that is representative of the measurements and the results re-evaluated." See change 1 above.

2. "It would be useful to have an indication of where the wetlands are in relation to the measurement site, as well as any landfills, and Fort McMurray (line 204), which is mentioned

but not included on Figure 1." We changed the area extent of the map to include the landfills and wetland approximate area in the figure. See the new Figure 1.

3. "It is unclear whether your results are consistent with the emissions inventories for the mining and tailings pond emissions. An explicit comment about this in your conclusion would be useful." This is not a straightforward comparison because the emission inventories available to the public report both tailings ponds and surface mine emissions in the category "fugitive $CH_4$ sources", and do not provide information about the proportional contribution of each of the sources comprising this category. The only real comparison we have is with the data of Baray et al., (2018).

4. "Emphasise the value of the carbon-14 measurements in your conclusions. Why is it important to use carbon-14 over, say, δD measurements with carbon-13?" The main value of carbon-14 is that it can trace $CH_4$ from fossil sources whether it is produced by microbial (as in tailings) or thermogenic processes. Other tracers can usually distinguish between biogenic and thermogenic processes but can not indicate if the substrate used by the microbes to produce $CH_4$ is fossil or not. We added a sentence saying this in line 290:

Line 290: While tracers such as $\delta^{13}C$, δD, and $C_2H_6/CH_4$ can separate thermally from microbially produced $CH_4$, the additional use of $\Delta^{14}C$ indicates if $CH_4$ is produced from a fossil source regardless of the process of $CH_4$ formation.

5. "The span of the sampling campaign is inconsistent in the manuscript. Line 96: "13-23 August" referring to the full campaign; Line 224 "20-24 August" referring to a subsample of measurements (end date different to line 96); Data presented in Table 1 and Figure 3 cover 16-23 August. You should check the length of the sampling campaign that covers the measurements presented is consistent throughout the paper to avoid confusion." We modified the text and Figure 2 to show the results from the 16 to the 23rd of August, that is when we performed the atmospheric measurements.

6. "The calibration problem with the G2201-i isotopic analyser (line 278) should be moved from your results to the methods section. If the data is not presented nor reliable it should not be mentioned in the research article." We removed this data from the results section as it is not used for the interpretation.

7. "Could you please clarify what you mean by 'simple linear regression' (line 157) Is this, for example, an ordinary least square or orthogonal distance regression? It would also be useful to know whether the measurement errors (shown in Figure 3) are included when calculating the linear regression parameters and if these uncertainties are included in the uncertainties of the y-intercept and gradient?" With simple we meant a linear regression with only one independent variable. We have updated this analysis as described above.

8. "On line 237 you state there is "one datapoint with a higher $\delta^{13}C$ than the background air". What is defined as the background air?" We changed this to say (line 246): However, there were only five data points, and four of them had very similar values which could artificially strengthen the correlation.

9. "It is stated differences between your study and Baray et al. (2018) is "mainly due to the uncertainty in the isotopic signatures of the $CH_4$ sources" (line 265). Are there any other main sources of uncertainty in your approach and the Baray et al. (2018) approach? Maybe consider the fetch of your measurements made at 3 metres? Is the 8-day sampling period representative of emissions in the AOSR?" The sampling is probably representative of summer only, the same as the estimates in Baray et al. (2018). Other differences between the two approaches that probably result in differences between source attributions include:

- The measurements in Baray et al. (2018) were done in 2013 and there were most likely changes in bitumen production (and therefore GHG emissions) in each mine since 2013. For example, in 2018 there was a decrease in oil sands production due to a change in governmental regulations.

- Airplane measurements were done in transects at heights from 150 to 1370 m above ground level, while ours were done at 3 m above ground level. However, they extrapolate the $CH_4$ mixing ratios to the surface levels in their results, which probably increases the uncertainty in their results, but it is unlikely that this is the main source of the differences between studies.

We changed line 263 to incorporate these ideas: We suggest that differences between studies could arise from changes in bitumen production in the different sites since 2013 and from the large uncertainties of our estimates.

10. "On line 279, in this instance p-values do not describe whether a relationship is linear. Please remove." We intended to show that this linear relationship is significant, but we will remove the p values as we are not showing these results.

11. "Figure 3. Please include the appropriate numerical symbols for the y-intercept on panels A, B, and C. Please explain the meaning of the y-intercept on panel C in the caption and text. Figure 3C is also not mentioned in the text, you should refer to it somewhere." We added the symbol ‰ in the y intercept, and a sentence in the caption saying the following: "In panels A and B, the intercept of the Keeling plot b indicated the isotopic signature of the $CH_4$ source. In panel C, the intercept b is interpreted as the $\delta^{13}C$ value of fossil $CH_4$". Figure 3C is mentioned in line 236.

12. "The labelling of Table 1 and Table 2 in the manuscript do not correspond to how these tables are presented. Please check the labelling." Table 1 and Table 2 labelling were corrected in the text.

13. "Table 2. Please include uncertainties for the concentration values. Please clarify if sample 11 was in fact made at 3:55 am (if so I admire your commitment) and that the $\Delta^{14}C$ value in sample 8 is that low." We added the uncertainties to the table (see the new Table 1). The sample was taken at 9:55 PM Mountain Time, but we report the time in UTC, hence the strange sampling times. We added a note to the caption of Table 1: "Note that local time of sampling (Mountain Time, MDT) is 6 hours behind UTC universal time". And yes, Sample 8 had quite a low $\Delta^{14}C$ value.

14. "I find it hard to interpret the spatial scale of Figure 2B. Adding a spatial scale bar or latitude-longitude values to the gridlines will make this clearer." We added a scalebar and labels to the gridline in Figure 2B (see new Figure 2B).

15. "On line 155, I would alter "simple linear mixing" to "simple mixing", linear doesn't make sense to me in this context." We changed the sentence according to the suggestion: "It assumes that the atmospheric CH$_4$ is the result of a simple mixing between two components"

16. "On line 23 (and throughout) what do mean by "10 ≤ 1 ‰"? This doesn't make sense mathematically." There is a ± symbol missing, the sentence should read: The results also indicate an influence of approximately 10 ± < 1% from microbial modern sources (Figure 4b), most likely from wetlands. We refer to the uncertainty in this way because the modeled uncertainty for wetland emissions is much less than for the other two source categories. This is because of the strong influence of the $^{14}$C measurements on the modeled wetland contribution.

17. "As per ACP guidelines latin phrases should not be in italics" We removed the italics in line 46: In contrast, the recovery of deeper deposits requires the use of in situ techniques that involve lowering the viscosity of bitumen by injecting steam into the reservoir to extract it (Bergerson et al., 2012).

18. "Miller et al. (2020) is missing from your list of references" The following reference was added to the list: Miller, J. B., Lehman, S. J., Verhulst, K. R., Miller, C. E., Duren, R. M., Yadav, V., Newman, S., and Sloop, C. D.: Large and seasonally varying biospheric CO2 fluxes in the Los Angeles megacity revealed by atmospheric radiocarbon, 117, 26681–26687, https://doi.org/10.1073/PNAS.2005253117/-/DCSUPPLEMENTAL, 2020.

19. "Units are presented using inconsistent notation. Please stick to index notation (as per ACP guidelines), leaving a space between the value and the unit (or symbol) and between different units e.g. 5 kg m$^{-2}$ s$^{-1}$ not 5kgm$^{-2}$s$^{-1}$" We changed the units to SI units (e.g. (line 114) 2000 PSI to 13.7 MPa, (section 2.1) 1 t ha$^{-1}$ a$^{-1}$ to 0.1 kg m$^{-2}$ a$^{-1}$) and added spaces between units.

20. "On line 109, What is th$^{-1}$?" It is tonnes (t) of CH$_4$ per hour. We have replaced it for the equivalent in kilograms to avoid confusion: (line 109) CH$_4$ emissions from CNRL Horizon facilities, Muskeg River and Jackpine Mines, and the Syncrude Aurora North Mine have been primarily attributed to open pit mining (5200 ± 1200 kg h$^{-1}$) …

21. "On line 217, you are missing the numerical symbol for value". We added the ‰ symbol to the numerical value. The sentence now reads: If we were to add a landfill component, assuming a $\delta^{13}$C value of -55 ‰ for landfills (Lopez et al., 2017) …

**References**

Baray, S., Darlington, A., Gordon, M., Hayden, K. L., Leithead, A., Li, S. M., Liu, P. S. K., Mittermeier, R. L., Moussa, S. G., O'Brien, J., Staebler, R., Wolde, M., Worthy, D., and McLaren, R.: Quantification of methane sources in the Athabasca Oil Sands Region of Alberta

by aircraft mass balance, Atmos. Chem. Phys., 18, 7361–7378, https://doi.org/10.5194/acp-18-7361-2018, 2018.

Graven, H., Hocking, T., & Zazzeri, G.: Detection of fossil and biogenic methane at regional scales using atmospheric radiocarbon, Earth'sFuture,7, 283–299. https://doi.org/10.1029/2018EF001064, 2019

Hammer, S., Levin, I.: Monthly mean atmospheric D14CO2 at Jungfraujoch and Schauinsland from 1986 to 2016, heiDATA, V2,https://doi.org/10.11588/data/10100, 2017

Stock, B. C., Jackson, A. L., Ward, E. J., Parnell, A. C., Phillips, D. L., and Semmens, B. X.: Analyzing mixing systems using a new generation of Bayesian tracer mixing models, PeerJ, 6, e5096, https://doi.org/10.7717/PEERJ.5096, 2018.

Turnbull, J. C., Fletcher, S. E. M., Ansell, I., Brailsford, G. W., Moss, R. C., Norris, M. W., and Steinkamp, K.: Sixty years of radiocarbon dioxide measurements at Wellington, New Zealand: 1954-2014, Atmos. Chem. Phys., 17, 14771–14784, https://doi.org/10.5194/ACP-17-14771-2017, 2017.

York, D., Evensen, N. M., Lopez Martinez, M., and De Basabe Delgado, J.: Unified equations for the slope, intercept, and standard errors of the best straight line, Am J. Phys., 72(3), 367–375, 2004.